# SARS-CoV-2 Renal Impairment in Critical Care: An Observational Study of 42 Cases (Kidney COVID)

**DOI:** 10.3390/jcm10081571

**Published:** 2021-04-08

**Authors:** Antoine-Marie Molina Barragan, Emmanuel Pardo, Pierre Galichon, Nicolas Hantala, Anne-Charlotte Gianinazzi, Lucie Darrivere, Eileen S. Tsai, Marc Garnier, Francis Bonnet, Fabienne Fieux, Franck Verdonk

**Affiliations:** 1Department of Anesthesiology and Intensive Care, Hôpital Saint-Antoine, Assistance Publique-Hôpitaux de Paris, 75012 Paris, France; antoine@molinabarragan.com (A.-M.M.B.); pardo.emmanuel@gmail.com (E.P.); nicolas.hantala@aphp.fr (N.H.); annecharlotte.gianinazzi@aphp.fr (A.-C.G.); lucie.darrivere@hotmail.fr (L.D.); marc.garnier@aphp.fr (M.G.); francis.bonnet@aphp.fr (F.B.); fabienne.fieux@aphp.fr (F.F.); 2Sorbonne University, GRC 29, DMU DREAM, Assistance Publique-Hôpitaux de Paris, 75013 Paris, France; pierre.galichon@aphp.fr; 3Transplantation and Nephrology Department, Hôpital Pitié-Salpétrière, Assistance Publique-Hôpitaux de Paris, 75013 Paris, France; 4Department of Anesthesiology, Perioperative and Pain Medicine, Stanford University School of Medicine, Stanford, CA 94305, USA; eileentsai425@gmail.com

**Keywords:** acute kidney injury, intrinsic renal injury, pneumonia, proteinuria, SARS-CoV-2

## Abstract

The severe acute respiratory syndrome coronavirus 2 (SARS-CoV-2) infection leads to 5% to 16% hospitalization in intensive care units (ICU) and is associated with 23% to 75% of kidney impairments, including acute kidney injury (AKI). The current work aims to precisely characterize the renal impairment associated to SARS-CoV-2 in ICU patients. Forty-two patients consecutively admitted to the ICU of a French university hospital who tested positive for SARS-CoV-2 between 25 March 2020, and 29 April 2020, were included and classified in categories according to their renal function. Complete renal profiles and evolution during ICU stay were fully characterized in 34 patients. Univariate analyses were performed to determine risk factors associated with AKI. In a second step, we conducted a logistic regression model with inverse probability of treatment weighting (IPTW) analyses to assess major comorbidities as predictors of AKI. Thirty-two patients (94.1%) met diagnostic criteria for intrinsic renal injury with a mixed pattern of tubular and glomerular injuries within the first week of ICU admission, which lasted upon discharge. During their ICU stay, 24 patients (57.1%) presented AKI which was associated with increased mortality (*p* = 0.007), hemodynamic failure (*p* = 0.022), and more altered clearance at hospital discharge (*p* = 0.001). AKI occurrence was associated with lower pH (*p* = 0.024), higher PaCO_2_ (CO_2_ partial pressure in the arterial blood) (*p* = 0.027), PEEP (positive end-expiratory pressure) (*p* = 0.027), procalcitonin (*p* = 0.015), and CRP (C-reactive protein) (*p* = 0.045) on ICU admission. AKI was found to be independently associated with chronic kidney disease (adjusted OR (odd ratio) 5.97 (2.1–19.69), *p* = 0.00149). Critical SARS-CoV-2 infection is associated with persistent intrinsic renal injury and AKI, which is a risk factor of mortality. Mechanical ventilation settings seem to be a critical factor of kidney impairment.

## 1. Introduction

About 5% to 16% [1,2] of the patients who tested positive for severe acute respiratory syndrome coronavirus 2 (SARS-CoV-2) required hospitalization in intensive care units (ICU), mainly for respiratory distress associating dyspnea, high respiratory rate, low oxygen saturation, or rapid increase in lung infiltrates [3,4]. Mortality associated with ICU hospitalizations ranged from 49% to 67% [5,6]. While respiratory symptoms are the cornerstone of the disease, other organs can be affected [7,8]. Acute kidney injury (AKI) occurred in 23% of SARS-CoV-2 patients during their hospitalization [9], and renal replacement therapy (RRT) was used for 13% of them [10]. However, when urine dipstick tests are systematically performed, the incidence of renal impairment on hospital admission can be evaluated as high as 75% [11]. The SARS-CoV-2 is thought to have a direct renal toxicity [12] via entry into proximal tubular cells and podocytes where angiotensin converting enzyme 2 (ACE2) receptors and transmembrane serine proteases (TMPRSS) are highly expressed [13,14,15]. In critically ill patients, other factors may be implicated, such as cytokine storm, angiotensin II pathway activation, dysregulation of complement, hypercoagulation, and microangiopathy [9]. Moreover, 14% of SARS-CoV-2 patients develop an acute respiratory distress syndrome (ARDS) [2] that is, by itself, also independently associated with AKI out of SARS-CoV-2 context [16,17]. This specific impairment could be associated with a higher mortality rate in ICU [18].

However, at this time, no study has characterized renal impairment of SARS-CoV-2 in ICU patients. Our work aims to define the intrinsic renal injury induced by SARS-CoV-2 infection in critically ill patients, its consequences in terms of renal function, and its relationship with morbidity and mortality.

## 2. Materials and Methods

### 2.1. Study Design and Patients

This single-center, observational study was conducted in a university hospital (Hôpital Saint-Antoine, Assistance Publique—Hôpitaux de Paris, France). Forty-two patients consecutively admitted to the ICU between 25 March 2020, and 29 April 2020, were included. ICU admission criteria were patient tested positive for SARS-CoV-2 by reverse transcription-polymerase chain reaction (RT-PCR) on nasopharyngeal or tracheal swab with clinical respiratory distress symptoms (polypnea, SpO_2_ < 90% with 5 L/min O_2_) and/or a PaO_2_/FiO_2_ ratio < 300 mm Hg. During their ICU stay, patients were routinely given urine and blood tests, which were analyzed afterwards. Patients were then classified in two groups: patients who presented AKI during their ICU stay and patients who did not. In both groups, urine profiles were used to detect signs of intrinsic renal injury or renal response to hypovolemia according to definitions below.

### 2.2. Definitions

The diagnosis of ARDS was carried out according to the Berlin criteria [19]. AKI scoring was defined according to the 2012 Kidney Disease: Improving Global Outcome (KDIGO) definitions [20] and AKI as KDIGO ≥ 1. We defined the serum creatinine baseline as the serum value on ICU admission, as previously described [21]. The glomerular filtration rate was calculated from the daily urine volume, creatininemia, and creatininuria. To determine whether patients had chronic kidney disease (CKD), we relied on their medical records or on the creatinine level of the last 6 months when it was present (12 patients). In order to characterize the renal impairment of patients, the following standard definitions were used: (i) renal response to hypovolemia—a marker predictive of prerenal acute kidney injury—as a fractional excretion of urea (FeUrea) of less than 35%, (ii) glomerular injury as the excess of high molecular weight proteins in urine: albuminuria > 0.03 g/24 h accounting for >50% of total. Samples with macroscopic hematuria were excluded as blood can interfere with the interpretation of proteinuria; (iii) proximal tubular injury as an increase in urinary low-molecular-weight proteins: urinary retinol binding protein (RBP) (>1.1 mg/24 h) or urinary alpha1-microglobulin (>15.8 mg/24 h) in the absence of glomerular injury (albuminuria < 50% daily proteinuria); (iv) renal tubular acidosis as a positive urinary anion gap in a context of metabolic acidosis; (v) mixed intrinsic kidney injury as the association of glomerular injury and renal tubular acidosis or proximal tubular injury with IgG > 11.3 mg/24 h in urine; glomerular injury, proximal tubular injury, renal tubular acidosis, and mixed intrinsic kidney injury are referred as intrinsic kidney injury. Augmented renal clearance (ARC) was defined as a clearance greater than 120 mL·min^−1^ [22].

### 2.3. Statistical Analysis

Mean and standard deviation (SD) or median and interquartile ranges (25th; 75th percentiles) were calculated for continuous variables, while numbers and percentages were calculated for categorical parameters. The normal distribution of each continuous variable was assessed with the use of the Shapiro–Wilk test. For the univariate analysis, categorical variables were compared between independent groups using the exact Fisher test or the Chi-square tests, and continuous variables were compared using the Student’s *t*-test or the Mann–Whitney test. Chord diagrams were used to depict the relationship between organic kidney impairment and kidney function at two different time points of the ICU stay.

In this observational study, risk of multivariate analysis bias was high. To address this potential bias, we conducted a logistic regression model with inverse probability of treatment weighting (IPTW) analyses to assess major comorbidities as predictors of AKI [23]. Weights for group with comorbidity (chronic kidney disease, chronic high blood pressure, or diabetes) were the inverse of propensity score and weights for group without were the inverse of 1—propensity score. We calculated propensity scores using multiple logistic regression analyses, with SARS-CoV-2 associated AKI as the dependent variable. Predictive covariates associated with selected comorbidities were chosen based on clinical judgment and model fit. The balance between the two groups was assessed as the standardized mean difference.

All statistical analyses were performed on R (version 3.6.2 for Macintosh, licenses GNU GPL, The R Foundation for Statistical Computing, Vienna, Austria). All tests were two-sided, and a *p*-value < 0.05 was considered for statistical significance.

### 2.4. Ethics and Approval and Consent to Participate

In accordance with the French law on biomedical research, this observational study obtained the approval of an Institutional Review Board (“Comité d’Éthique de la Recherche en Anesthésie-Réanimation” (CERAR, President Prof. JE Bazin, 05/30/20) under the reference IRB 00010254-2020-106) [24]. In order to guarantee the security of personal data, the investigators retrospectively collected and integrated the information anonymously into a secure database in accordance with the French Commission Nationale de l’Informatique et des Libertés (CNIL) reference methodology (MR)—004 and registered it in the AP-HP processing register under number 20200803123416.

## 3. Results

### 3.1. Characteristics of the Study Population

During the study period, between 25 March 2020, and 29 April 2020, 42 patients were admitted in ICU for SARS-CoV-2 infection. The characteristics of the population are summarized in Table 1. Patients, median age 61.5 years old (interquartile range (IQR) (54.2; 65 years)) were hospitalized in ICU on a median of 8 days (IQR,7;12 days) after their first SARS-CoV-2 symptoms. Among them, 25 patients (59.5%) presented at least one comorbidity, including CKD in seven patients (16.7%), and overweight with a median body mass index (BMI) of 27.2 kg·m^−2^ (IQR, 24.3 kg·m^−2^; 30 kg·m^−2^). On ICU admission, most of the patients presented a moderate to severe ARDS (41 out of 42) with a median ratio of arterial partial pressure of oxygen and inspired fraction of oxygen (PaO_2_/FiO_2_) of 140 (IQR 103.7; 172.9). Twenty-six patients (61.90%) had a CT-scan at ICU admission and patients were scored using the European Society of Radiology score. This score graded the parenchymal involvement according to a visual classification in five stages based on the percentage of lung damaged [25]. Ten patients (23.8%) presented an altered clearance (<60 mL·min^−1^) at their ICU admission. See Appendix A.

### 3.2. Kidney Abnormalities during ICU Stay

The first blood analysis including inflammatory cytokines and urinary assessment was performed on a median of 8 days after ICU admission (IQR 6; 10 days) (Table 2), see Appendix A, table with biological values during the ICU stay. Complete urinary samples allowing one to define intrinsic renal injury were available for 34 patients. Eight patients were excluded from the analysis because no urinary profiles were completed during ICU stay (retinol binding protein, alpha1-microglobulin, albuminuria, and proteinuria) due to technical problems.

Ten patients (29.4%) had AKI according to KDIGO criteria at the time of urinary profile collection (four patients were KDIGO1, one was KDIGO2, and five were KDIGO3). ARC was documented in nine patients (31%).

Fifteen patients (48.4%) presented criteria for renal response to hypovolemia, and 32 patients (94.1%) met the diagnostic criteria for intrinsic kidney injury. Among the latter, 26 patients (76.5%) had mixed intrinsic kidney injury, while six (17.6%) had a proximal tubular injury, and none had glomerular injury. Twenty-three patients had significant albuminuria (defined as albuminuria > 0.03 g/24 h).

### 3.3. Kidney Abnormalities on ICU Discharge

The second blood analysis including inflammatory cytokines and urinary assessment was performed on discharge from ICU at a median of 20 days after admission (IQR 15.75; 23.25 days) (Table 3). See Appendix A, table with biological values on ICU discharge. Urinary samples were available for 16 patients who stayed in ICU, eight were dead, and 10 were transferred to another hospital without any urinary analysis.

On discharge, 4 patients (25%) had AKI according to KDIGO criteria (two patients were KDIGO 1, one patient was KDIGO 2, and one was KDIGO 3). Conversely, ARC was found in six patients (50%).

Five patients (35.7%) presented criteria for renal response to hypovolemia, and 16 patients (100%) met the diagnostic criteria for intrinsic kidney injury. Among them, mixed injury was documented in 12 patients (75%), while proximal tubular injury was documented in four patients (25%). Seven patients had significant albuminuria.

Relationship between kidney function estimated by KDIGO and intrinsic kidney injury at early and late stages are represented in Figure 1.

### 3.4. Prognosis Associated with AKI

Twenty-four patients (57.1%) presented AKI during their ICU stay. Mortality was higher in AKI patients (33.3% vs. 0%, respectively, in AKI and non-AKI groups, *p* = 0.007) (Table 4). The number of patients who received norepinephrine during their stay did not differ between groups; however, the duration of catecholamine use was significantly longer in the AKI group (7 days vs. 2 days, respectively, in AKI and non-AKI groups, *p* = 0.022). Infections and septic shock during intensive care hospitalization were significantly more common in the AKI group (45.8% vs. 11.1, respectively, in AKI and non-AKI groups, *p* = 0.021). Conversely, length of stay in ICU (20 days vs. 19.5, respectively, in AKI and non-AKI groups, *p* = 0.507) and invasive mechanical ventilation duration (22 days vs. 17 days, respectively, in AKI and non-AKI groups, *p* = 0.173) were comparable between the two groups. Discharge from hospital occurred at a median of 30 days after admission (IQR 18.5; 46.5 days). At this time, patients with AKI during ICU stay had a lower value of creatinine clearance (37.93 mL·min^−1^ vs. 121.04 mL·min^−1^, respectively, in AKI and non-AKI groups, *p* = 0.005). Creatinine clearance significantly decreased from hospital admission to discharge when AKI occurred during the stay.

ARC was observed in 23 patients (54.8%) and more frequently in the non-AKI group (29.2% and 88.9%, respectively, in AKI and non-AKI groups, *p* < 0.001). See Appendix A.

On univariate analysis, patients of the AKI group had higher creatinine levels on hospital admission (94 µmol·L^−1^ vs. 74 µmol·L^−1^, *p* = 0.037). They also had higher BMI (29.4 kg·m^−2^ vs. 25.1 kg·m^−2^, *p* = 0.009), higher PaCO_2_ (42 mm Hg vs. 35 mm Hg, *p* = 0.027), lower pH (7.35 vs. 7.44, *p* = 0.024), higher positive end-expiratory pressure (PEEP) (12 cm H_2_O vs. 8 cm H_2_O, *p* = 0.027), higher procalcitonin (PCT) blood concentration (1.56 ng·mL^−1^ vs. 0.27 ng·mL^−1^, *p* = 0.015), and higher C-reactive protein (CRP) blood concentration (274.3 mg·L^−1^ vs. 167 mg·L^−1^, *p* = 0.045) on ICU admission in comparison to non-AKI group (Table 1 and Appendix A).

Other known risk factors such as nephrotoxic agents’ infusion (e.g., contrast agents, diuretics, and aminosides), negative fluid balance, and admission severity scores (SOFA and SAPSII scores) did not show any statistical association with the occurrence of AKI.

In the weighted logistic regression analysis (Table 5), CKD was found to be independently associated with AKI (adjusted OR (odd ratio) 5.97 (2.1–19.69), *p*-value = 0.00149). However, no significant association was found regarding diabetes or chronic high blood pressure.

## 4. Discussion

The current cohort is representative of critical SARS-CoV-2 patients with 80% of overweight men with a median age of 61.5 years and at least one comorbidity (including hypertension and diabetes mellitus) [26,27]. All of the patients had invasive mechanical ventilation because of moderate to severe ARDS onset, which is in line with previous retrospective studies [26]. Patients were comparable in terms of severity (median SOFA score of 7) to those usually admitted for ARDS [28].

Within the first week of ICU admission, 94.1% of patients had features of intrinsic kidney injury and all the patients had documented intrinsic kidney injury three weeks after admission. Fifty-seven percent of patients presented AKI according to KDIGO ranking, and one third of them required RRT (Table 4). Conversely, 54.8% of patients presented ARC during their ICU stay including 29.2% patients in the AKI group. Intrinsic renal injury remains stable for both samples and is mixed (76.5% during ICU stay, 75% on discharge), and proximal tubular (17.6% during ICU stay and 25% on discharge). To our knowledge, this is the first study describing with this level of precision kidney injury during SARS-CoV-2 infection leading to ICU care.

In our cohort, 82.5% of critical SARS-CoV-2 patients presented proteinuria, indicating a high rate of intrinsic kidney injury which is in line with Pei et al. concerning the Wuhan pandemic [11]. In ICU patients, acute tubular necrosis is well documented up to 78% of patients in the autopsy series [29], whereas this current population of critically ill SARS-CoV-2 patients presented mixed-pattern lesions [29,30]. The prognostic significance of proteinuria or intrinsic kidney injury is unclear: the high prevalence of abnormal proteinuria shows that the majority of patients were exposed to a kidney stress, even in those who did not develop bona fide AKI (subclinical AKI). Subclinical AKI can denote either an adequate coping with an acute stress or an early stage of a progressive chronic kidney disease. Careful follow up of these patients is warranted to determine the long-term consequences of this acute episode. It might also reflect a specific effect of SARS-CoV-2 infection on kidney cells, as suggested by previous descriptions of viral inclusions and by the expression of the ACE2 (the virus’ putative receptor for cell invasion) in renal glomerular and tubular cells [9]. It has to be noted that all the patients had a urinary catheter that could induce proteinuria in case of traumatic catheterization, reinforcing the importance of a complete urinary profile to describe the intrinsic kidney injury specific to SARS-CoV-2. In another study conducted by Cheng et al. [21], including ICU and non-ICU patients, only 9.8% of the patients had a urinary catheter, whereas 43.9% had proteinuria. This supports the idea that SARS-CoV-2 was responsible for the proteinuria that could be as high as 6.6 g/L in the current cohort.

In the current cohort, 94.12% patients presented with either AKI (KDIGO ≥ 1) or intrinsic renal impairment (tubular or glomerular or mixed injury with KDIGO = 0). Abnormal levels of proteinuria in the non-AKI group indicates subclinical AKI, and further increases in proteinuria in the overt AKI group suggest that both groups undergo a kidney injury, with a continuum of severity between both groups. On the other hand, we assessed the contribution of prerenal mechanisms to AKI in these patients using the fractional excretion of urea (FeUrea). FeUrea is a robust marker of the renal response to hypovolemia more relevant than other markers like fractional excretion of sodium or the urine-to-plasma ratio of creatinine, especially in case of diuretic use. According to FeUrea values, hypovolemia is frequent among SARS-CoV-2 patients in ICU, but the lack of association between AKI and low FeUrea values strengthens the role of intrinsic kidney injury associated to SARS-CoV-2 infection in AKI occurrence.

Certain factors may have contributed to the development of AKI. Norepinephrine was used in 33 patients (80.5%), which is consistent with the data in the literature ranging from 35 to 94% depending on the series [31]. Only one patient received dobutamine in the context of septic cardiomyopathy. However, the duration of catecholamine use was greater in the AKI group, as was the proportion of septic shock. The greater susceptibility of AKI patients to infection is demonstrated in the literature [32] and is well reflected in our cohort with 87.5% and 38.89% of infection during ICU stay, respectively, in the AKI and non-AKI group, and 45.83% and 11.11% of septic shock, respectively, in the AKI and non-AKI group.

Although the small number of patients did not allow one to perform a multivariate analysis, univariate analysis highlighted several factors that differ between AKI and non-AKI patients during SARS-CoV-2 infection. Firstly, prehospital kidney function, defined as renal functional reserve [33], appears to be a critical factor in renal prognosis, as CKD is independently associated with AKI occurrence (5.97 (2.1–19.69), *p* = 0.00149)). This renal functional reserve depends on patient’s prior renal function but also on many other factors: hypovolemia induced by prolonged fever, infection-induced digestive disorders [34], and chronic hypertension, especially if treated by ACE inhibitors [35,36]. However, hypovolemia may, at least partly, play a role, as up to 50% of SARS-COV-2 patients had modified fractional excretion of urea during their ICU stay related to ARDS management, which requires avoidance of excess fluid infusion [37].

Secondly, in the current cohort, all the patients presented high inflammatory response to SARS-CoV-2 infection, including high levels of IL6 plasma concentrations, as documented in previous studies [38], and moreover, those who developed AKI had significantly higher CRP and PCT values on ICU admission.

Thirdly, mechanical respiratory support including the level of PEEP is critical in developing AKI in SARS-CoV-2. As described by Hirsch et al., SARS-CoV-2 ventilated patients are at greater risk of developing AKI than non-ventilated ones (89.7% vs. 21.7%), especially during the first 24 h following intubation [39]. Elevation of central venous pressure due to high intrathoracic pressures may result in an increased kidney hydrostatic pressure, which leads to glomerular filtration impairment [40,41]. Patients in both groups (AKI or non-AKI) presented similar PaO2/FiO2 ratios, indicating that the severity of the lung impairment is independent of kidney injury and, thus, reinforcing the importance of a tightly controlled and tailored PEEP level. Finally, even if univariate analysis did not show any association between the type of organic impairment and AKI, all critical patients presented intrinsic kidney injury 3 weeks after admission, which should contribute to the loss of renal functional reserve.

Renal prognosis is also a key issue for SARS-CoV-2 patients, considering the incidence of intrinsic kidney injury and the proportion of AKI. In a study including 5273 non-SARS-CoV-2 patients with no pre-existing CKD, who developed AKI during an ICU stay, de novo AKI was associated with increased short- and long-term risk of death at 1 and 5 years [42]. At 1 year, AKI acquired in ICU was also independently associated with increased risk of CKD (6%) and end-stage renal disease (2%). In our study, among the 42 patients, 24 patients (57.1%) presented AKI during their ICU stay. When discharged from the hospital, 45.8% of the patients who developed AKI had impaired renal function with a creatinine clearance < 60 mL·min^−1^. Given the facts that patients had several known risk factors for developing chronic kidney disease (sepsis, ARDS, mechanical ventilation, and inflammatory and procoagulant responses) and that direct viral injury on kidney cells is still not yet fully characterized, long-term nephrological follow-up may be required for all those who have developed AKI, especially if they have other comorbidities or still have proteinuria.

The current study also revealed ARC in more than 50% of patients, especially in those who did not present AKI during their ICU stay (88.9%). In a systematic review, 20 to 65% of critically ill patients presented ARC with a higher prevalence in trauma patients [43]. Two main mechanisms have been suggested to explain ARC: (i) the release of pro-inflammatory cytokines that would lead to a decrease in resistance and an increase in glomerular filtration rate thanks to an increase in cardiac output and (ii) the efficacy of the physiological renal reserve allowing an increased glomerular filtration rate to cope with certain pathological situations such as ICU care. However, we did not find an increase in albuminuria in patients with ARC, suggesting that it more likely reflects just a hemodynamic adaptation. According to some authors, ARC is a good prognostic factor, which is confirmed by our study [44]. This ARC highlights a potential issue in critical patients because the use of regular doses of renally cleared drugs might induce underdosage [45]. This is of most importance considering antimicrobial treatment such as beta-lactam antibiotics, vancomycin, or aminoglycosides, where ARC may condition clinical failure or emergence of resistance if higher dosage is not used.

Eventually, another point highlighted by the current study is the significant association between kidney failure and mortality, as 33.3% of AKI patients died in comparison to 0% in the non-AKI patients. This relationship was suggested in previous studies in the pandemic context [21] with a 3.5-fold higher mortality in the case of AKI KDIGO stage 2 or more [46].

Finally, there are no data in the literature to compare the urinary profiles of our cohort’s SARS-CoV-2 patients with non-SARS-CoV-2 ARDS. The proportion of AKI in our study (57.14%) corresponds to the data in the literature (29.5% to 68.3% depending on the series) [16,41,47]. Nevertheless, the important prevalence of glomerular involvement suggests the participation of SARS-CoV-2 specific mechanisms as glomerular lesions are not classically described in ARDS.

This study has several limitations. First, it includes only a limited number of patients (42 patients) in one university hospital, making it impossible to rule out residual confusion and bias. In addition, some clinical and biological data from the admission to discharge from ICU were missing. Furthermore, we used the baseline creatinine level at hospital entry as the baseline creatinine level for some patients, which could lead to an underestimation of AKI. Moreover, CKD was assessed on medical records or on the creatinine level of the last 6 months obtained for 12 patients, which could also lead to an underestimation. Finally, blood test and urine samples were made at the discretion of each physician depending patient’s evolution, leading to variations in the time of the first complete renal function evaluation.

## 5. Conclusions

The prevalence of SARS-CoV-2 intrinsic kidney injury in critical care patients is high. A mixed injury is noticed early and persists during hospital stay. AKI is associated with mortality in excess in ICU patients and poor renal outcome. Detecting intrinsic renal injuries by analyzing complete urinary profiles on ICU admission might be recommended to adapt the clinical management of critical SARS-CoV-2 patients.

## Figures and Tables

**Figure 1 jcm-10-01571-f001:**
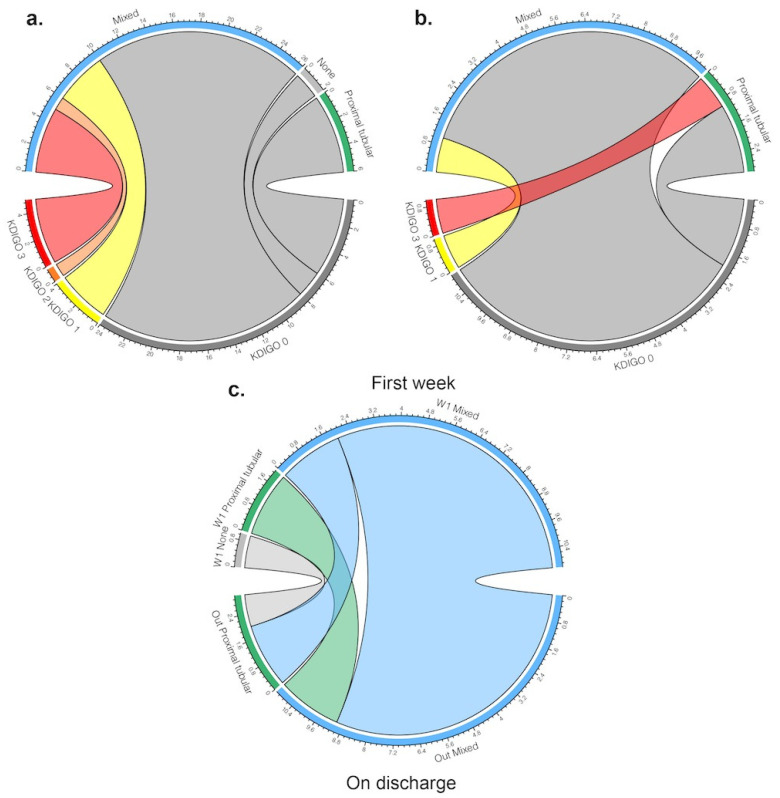
Chord diagrams representing characteristics of kidney injury during the ICU stay. (**a**) Relationship between kidney function estimated by the 2012 Kidney Disease: Improving Global Outcome (KDIGO) and intrinsic kidney injury within the first week after ICU admission; (**b**) Relationship between kidney function estimated by KDIGO and intrinsic kidney injury on ICU discharge; (**c**) Evolution of the intrinsic kidney injury between the first week after ICU admission and ICU discharge. The bottom part of the diagram represents patients sorted by their KDIGO classification, and the top part represents the same patients ranked according to the intrinsic kidney injury diagnosis made by profiling urinary analysis. Ribbons show for every patient the connection between kidney injury and function.

**Table 1 jcm-10-01571-t001:** Baseline and ICU admission patient’s characteristics.

	Overall (*n* = 42)	Non-AKI (*n* = 18)	AKI (*n* = 24)	*p*-Value
Demographic and Clinical Data				
Age (years)	61.50 (54.25, 65.00)	60.50 (49.75, 66.00)	61.50 (55.50, 65.00)	0.684
Male (%)	34 (81.0)	15 (83.3)	19 (79.2)	1.000
BMI (kg·m^−2^)	27.25 (24.30, 30.00)	25.10 (23.80, 27.10)	29.40 (27.40, 30.90)	0.009
No comorbidities (%)	17 (40.5)	7 (38.9)	10 (41.7)	1.000
CKD (%)	7 (16.7)	2 (11.1)	5 (20.8)	0.679
SOFA score	7.00 (4.00, 9.75)	4.50 (3.00, 8.50)	8.00 (6.00, 11.25)	0.134
Biological Data				
CRP (mg·L^−1^)	246.15(144.88, 300.20)	167.00 (140.52, 275.75)	274.30 (213.85, 330.12)	0.045
Procalcitonin (ng·mL^−1^)	1.01 (0.29, 2.50)	0.27 (0.13, 0.91)	1.56 (0.62, 3.70)	0.015
Confirmed concurrent infection	12 (28.6)	1 (5.6)	11 (45.8)	0.005
GFR by MDRD (mL·min^−1^)	87.72 (64.71, 117.29)	91.80 (72.38, 135.85)	82.47 (54.53, 102.27)	0.213
Creatinine (µmol·L^−1^)	76.50 (61.75, 100.75)	74.50 (54.25, 93.50)	82.00 (69.25, 118.50)	0.208
ARC (%)	10 (23.8)	6 (33.3)	4 (16.7)	0.281
pH	7.41 (7.34, 7.47)	7.44 (7.40, 7.48)	7.35 (7.29, 7.44)	0.024
PEEP (cm H_2_O)	12.00 (8.25, 12.00)	8.00 (8.00, 12.00)	12.00 (10.00, 14.00)	0.027
PaCO_2_ (mm Hg)	40.00 (34.00, 44.00)	35.00 (32.25, 40.75)	42.00 (35.00, 47.50)	0.027
PaO_2_/FiO_2_ Ratio	140.00 (103.75, 172.86)	112.50 (95.75, 148.25)	148.00 (113.12, 182.86)	0.124
Radiological Characteristic				
European Society of Radiology score				
Grade 1	1 (3.8)	0 (0.0)	1 (6.7)	0.958
Grade 2	5 (19.2)	2 (18.2)	3 (20.0)	
Grade 3	7 (26.9)	4 (36.4)	3 (20.0)	
Grade 4	11 (42.3)	4 (36.4)	7 (46.7)	
Grade 5	2 (7.7)	1 (9.1)	1 (6.7)	

Values are expressed as median (interquartile ranges), absolute value (percentages); ARC (augmented renal clearance), BMI (body mass index), CKD (chronic kidney disease), CRP (C-reactive protein), GFR (glomerular filtration rate), PaCO_2_ (CO_2_ partial pressure in the arterial blood), PEEP (positive end-expiratory pressure), SOFA (sequential organ failure assessment score), ICU: intensive care units; AKI: acute kidney injury.

**Table 2 jcm-10-01571-t002:** Kidney function and impairment characterization during ICU stay.

	Overall (*n* = 34)	Non-AKI (*n* = 15)	AKI (*n* = 19)	*p*-Value
Days from ICU admission	8.00 (6.00, 10.00)	8.00 (6.00, 10.00)	8.00 (5.50, 10.00)	0.958
Creatinine clearance (mL·min^−1^)	96.21 (67.69, 126.41)	126.41 (105.48, 157.48)	68.73 (41.52, 86.17)	<0.001
ARC (%)	9 (31.0)	7 (53.8)	2 (12.5)	0.041
KDIGO (%)				0.002
0	24 (70.6)	15 (100.0)	9 (47.4)	
1	4 (11.8)	0 (0.0)	4 (21.1)	
2	1 (2.9)	0 (0.0)	1 (5.3)	
3	5 (14.7)	0 (0.0)	5 (26.3)	
Renal Impairment				
Hypovolemia (%)	15 (48.4)	8 (57.1)	7 (41.2)	0.479
Intrinsic kidney injury (%)				0.211
Glomerular	0 (0.0)	0 (0.0)	0 (0.0)	
Mixed	26 (76.5)	10 (66.7)	16 (84.2)	
Proximal tubular	6 (17.6)	3 (20.0)	3 (15.8)	
Tubular acidosis	0 (0.0)	0 (0.0)	0 (0.0)	
None	2 (5.9)	2 (13.3)	0 (0.0)	

Values are expressed as median (interquartile ranges), absolute value (percentages); ICU: intensive care unit), ARC: augmented renal clearance.

**Table 3 jcm-10-01571-t003:** Kidney abnormalities on ICU discharge.

	Overall (*n* = 16)	Non-AKI (*n* = 7)	AKI (*n* = 9)	*p*-Value
Days from ICU admission	20.00 (15.75, 23.25)	20.00 (15.00, 22.50)	20.00 (19.00, 23.00)	0.749
Creatinine clearance (mL·min^−1^)	113.49 (69.30, 142.07)	139.90 (81.20, 143.75)	101.11 (50.37, 133.69)	0.372
ARC (%)	6 (50.0)	3 (60.0)	3 (42.9)	1.000
KDIGO (%)				0.070
0	10 (71.4)	7 (100.0)	3 (42.9)	
1	2 (14.3)	0 (0.0)	2 (28.6)	
2	1 (7.1)	0 (0.0)	1 (14.3)	
3	1 (7.1)	0 (0.0)	1 (14.3)	
Renal Impairment				
Hypovolemia (%)	5 (35.7)	2 (28.6)	3 (42.9)	1.000
Intrinsic kidney injury (%)				1.000
Glomerular	0 (0.0)	0 (0.0)	0 (0.0)	
Mixed	12 (75.0)	5 (71.4)	7 (77.8)	
Proximal tubular	4 (25.0)	2 (28.6)	2 (22.2)	
Tubular acidosis	0 (0.0)	0 (0.0)	0 (0.0)	
None	0 (0.0)	0 (0.0)	0 (0.0)	

Values are expressed as median (interquartile ranges), absolute value (percentages); ARC: acute renal clearance; ICU: intensive care unit.

**Table 4 jcm-10-01571-t004:** Prognosis of severe acute respiratory syndrome coronavirus 2 (SARS-CoV-2) patients according to the presence of an acute kidney injury during the ICU stay.

	Overall (*n* = 42)	Non-AKI (*n* = 18)	AKI (*n* = 24)	*p*-Value
Kidney Function during ICU Stay				
ARC (%)	23 (54.8)	16 (88.9)	7 (29.2)	<0.001
KDIGO				
KDIGO 0 (%)	18 (42.9)	18 (100.0)	0 (0.0)	<0.001
KDIGO 1 (%)	12 (28.6)	0 (0.0)	12 (50.0)	
KDIGO 2 (%)	3 (7.1)	0 (0.0)	3 (12.5)	
KDIGO 3 (%)	9 (21.4)	0 (0.0)	9 (37.5)	
Organ Support during ICU Stay				
Length of mechanical ventilation (days)	19.00 (11.00, 28.00)	17.00 (6.25, 23.75)	22.00 (12.00, 34.00)	0.173
Norepinephrine (patients)	33 (80.5)	14 (77.8)	19 (82.6)	0.713
Dobutamine	1 (2.4)	0 (0.0)	1 (4.2)	1.000
Catecholamine duration (days)	5.00 (1.00, 10.00)	2.00 (1.00, 5.50)	7.00 (3.50, 14.50)	0.022
Dialysis (%)	9 (21.4)	0 (0.0)	9 (37.5)	0.005
Septic shock	13 (31.0)	2 (11.1)	11 (45.8)	0.021
Infection during ICU stay	28 (66.7)	7 (38.9)	21 (87.5)	0.002
Prognosis				
Creatinine clearance < 60 mL·min^−1^ on ICU discharge (%)	11 (27.5)	0 (0.0)	11 (45.8)	0.001
ICU LOS (days)	19.50 (14.00, 33.25)	20.00 (13.75, 30.75)	19.50 (15.50, 36.25)	0.507
ICU mortality (%)	8 (19.0)	0 (0.0)	8 (33.3)	0.007

Values are expressed as median (interquartile ranges), absolute value (percentages); ARC (acute renal clearance), LOS (length of stay).

**Table 5 jcm-10-01571-t005:** Comparison of the incidence of acute kidney injury (AKI) according to major comorbidities.

Outcome	Factor	Crude		IPTW	
		OR (95% CI)	*p*-Value	OR (95% CI)	*p*-Value *
AKI	CKD	2.11 (0.39–16.11)	0.41	5.97 (2.1–19.69)	0.00149
	Diabetes	1.17 (0.28–5.32)	0.834	0.86 (0.36–2.01)	0.722
	Chronic high blood pressure	1.06 (0.31–3.67)	0.929	0.75 (0.32–1.76)	0.514

IPTW: inverse probability of treatment weighting, CKD: chronic kidney disease, OR: odd ratio. *p*-value * by weighted logistic regression.

## Data Availability

The results presented in this paper have not been published previously in whole or part, except in abstract format. The datasets used and analyzed during the current study are available from the corresponding author on reasonable request.

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
