# Peer review of "SARS-CoV-2 Renal Impairment in Critical Care: An Observational Study of 42 Cases (Kidney COVID)"

_jcm, 2021, doi:10.3390/jcm10081571_

Round 1
Reviewer 1 Report
The authors have made a good revision of great interest andhigh originality.
Author Response
We would like to thank the reviewer for his prompt revision and his comment.
Reviewer 2 Report
Thanks to the authors for addressing my comments. I have no further concerns.
Author Response
We would like to thank the reviewer for his prompt revision and his comment.
This manuscript is a resubmission of an earlier submission. The following is a list of the peer review reports and author responses from that submission.
Round 1
Reviewer 1 Report
This is a very well researched, referenced and written paper. I appreciate the chance to review but have no suggestions on any improvements that should be made.
Reviewer 2 Report
The authors presented the results of a single center observational study about the incidence of AKI, response to hypovolemia and intrinsic renal injury at the admission to and discharge from ICU. The sample size, as already noted by the authors, represents the main limitation of the study.
Major revisions
- Overlapping definition: Methods (page 2 lines 77-80: the definition of glomerular injury seems the same of that of mixed intrinsic kidney injury at page 2 line 86-87 "...the presence of urinary IGG (>11.3 mg/24h) associated with albuminuria < 50% of daily proteinuria".
- Catecolamine duration is significantly different between AKI and non-AKI group (page 6 lines 175) 7 days vs 2 days: How many patients needed catecolamine? Which catecolamine has been used? How many patients experienced septic shock or cardiogenic shock? These are all conditions which influence kidney function? if possible, provide information about the timing of AKI and the start i.e. how many patients developed AKI after starting catecolamine: this is important to determine if hypotension affected renal function and, regarding the type of catecolamine used, it is important to determine the renal vasoconstriction.
- Page 7 lines 190-197: AKI group seemed to experienced a significant worst clinical condition and even if P/F ratio is not significantly different between AKI and non-AKI group, all the other parameters revealed a more severe respiratory condition (PEEP and PaCO2); P/F score, on the other hand, is strongly influenced by shunt and, in these cases, varies at different FiO2 and it is not a good parameter to compare the severity of clnical condition of different patients (page 8 lines265-267). Moreover, it would be interesting to know if radiographic findings on CT scan are significantly different between the groups in terms of chest score. Moreover, the level of procalcitonin is significantly higher in AKI group: there is no information about the difference in the two group of superimposed infections and it seems that AKI group experienced more bacterial infections which can complicate and influence renal condition.
- Page 7 lines 204-207. Please specify if invasive or non invasive mechanical ventilation.
- One limitation that could be underlined is that the first blood and urine analysis was performed after a median of 8 days after ICU admission and that this admission happened about 8 days after first symptoms: it means that the first assessment was made after about 16 days after the onset of symptoms: it is a long time in which the authors does not know the evolution of renal markers. Why blood and urine samples were taken after 8 days after ICU admission?
- Page 8 line 223 Please explain why proteinuria and intrinsic kidney injury "might reflect a strong adaptative potential of kidney".
- Pge 8 line 235-237: "Lower molecular weight proteins excretion (such as alpha1-microglobulin) in the non-AKI group in comparison to the AKI group suggests a continuum from intrinsic kidney injury to AKI." In our opinion this continuum should be suggested by the evolution in the same group from a condition with Lower molecular weight proteins excretion to a straight AKI condition and not by the comparison of two group.
- It would be important to compare the frequency of AKI, response to hypovolemia and intrinsic renal injury between your population and litterature data of a non-COVID ARDS population to underline the peculiar characteristics of renal impairiment in SARS-COV2 infection. Readers could be doubtful if these are findings related to ARDS or specifically to COVID condition. Renal impairment in such a disease as moderate and severe ARDS could be so strong (i.e. for the systemic inflammatory involvement) that the specific damage due to SARS-COV2 could be masked.
- Page 8 line 246-247; line 268-269: univariate analysis cannot allow naither to establish any association between variables nor to drag any conclusion on predictive factors.
Minor revisions
- Augmented renal clearance is always written in extenso and not with the acronym ARC used for the first time at line page 2 line 89.
- Page line 228: Traumatic catheterization can explain a prolonged proteinuria? I supposed that patients received urina catheter at the admission and the first urine sample was taken 8 days after.
- Page 1 line 39: please cite these works to link references about other organs involvement in COVID 19:
Thakur V, Ratho RK, Kumar P, Bhatia SK, Bora I, Mohi GK, Saxena SK, Devi M, Yadav D, Mehariya S. Multi-Organ Involvement in COVID-19: Beyond Pulmonary Manifestations. J Clin Med. 2021 Jan 24;10(3):446. doi: 10.3390/jcm10030446. PMID: 33498861; PMCID: PMC7866189.
Manganelli F, Vargas M, Iovino A, Iacovazzo C, Santoro L, Servillo G. Brainstem involvement and respiratory failure in COVID-19. Neurol Sci. 2020 Jul;41(7):1663-1665. doi: 10.1007/s10072-020-04487-2. Epub 2020 May 29. PMID: 32472516; PMCID: PMC7256922.
4. Page 2 line 46: Please cite: "Capuano I, Buonanno P, Riccio E, Pisani A. Acute Kidney Injury in COVID-19 Pandemic. Nephron. 2020;144(7):345-346. doi: 10.1159/000508381. Epub 2020 May 19. PMID: 32428921; PMCID: PMC7270059." where there is an insight of renal tropism of SARS-CoV2 MERS-CoV, and SARS-CoV, the receptors implicated and the potential explanation of different renal involvement in these three correlated viruses.
Reviewer 3 Report
"SARS-CoV2 renal impairment in Critical Care: an observational study of 42 cases (Kidney COVID)" is a critical and timely manuscript. However, the current version of the manuscript has some limitations, and if authors can address the same and make the suggested changes, it would benefit the manuscript.
Major comments:
- In the discussion (lines 245-247), the authors mention that the small sample size did not allow them to perform multivariate analysis. However, authors can use inverse probability treatment weighting (IPTW) to estimate the weights based on available information and conduct an unadjusted logistic regression followed by an adjusted logistic regression (adjusted for weights). It would benefit the manuscript to perform an analysis with renal function reserve as a predictor adjusted for weights.
- I would also recommend the authors perform a similar analysis changing the predictor to hypertension and diabetes, respectively.
- Authors can also perform a weight-adjusted analysis where AKI is the predictor/exposure and mortality as the outcome.
Adding multivariate analysis would strengthen the robustness of the finding observed in this manuscript.